# A New Diterpene with Cytotoxic Potential Against Human Tumor Cells

**DOI:** 10.3390/molecules30234629

**Published:** 2025-12-02

**Authors:** Orfa Inés Contreras-Martínez, Briana Alarcón Avilés, Fillipe Vieira Rocha, Karine Zanotti, Tamara Teixeira, Jesus Sierra Martínez, Alberto Angulo-Ortíz

**Affiliations:** 1Biology Department, Faculty of Basic Sciences, University of Córdoba, Montería 230002, Colombia; oicontreras@correo.unicordoba.edu.co; 2Chemistry Department, Faculty of Basic Sciences, University of Córdoba, Montería 230002, Colombia; bcalarconaviles@correo.unicordoba.edu.co; 3Chemistry Department, Federal University of São Carlos, São Carlos 13565-905, Brazil; fillipe@ufscar.br (F.V.R.); karinezanotti_@hotmail.com (K.Z.); tamara.teixeira.296@gmail.com (T.T.); 4Genetics and Evolution Department, Federal University of São Carlos, São Carlos 13565-905, Brazil; jesussierra@estudante.ufscar.br

**Keywords:** biisoespintanol, diterpene, cytotoxicity, tumor cell lines

## Abstract

Cancer is one of the most feared diseases in the world. Its incidence has increased steadily in recent years; it represents a significant burden of disease and is among the leading causes of death globally. Consequently, the search for novel compounds that serve as potential candidates for pharmacotherapeutic options and that can be used as treatments or adjuvants to control this disease is urgent. In this context, plant-derived phenolic diterpenes have shown antitumor activity against several types of cancer, inhibiting DNA synthesis, lipid metabolism, and bioenergetics of these cells, among other mechanisms, making these compounds an excellent alternative to continue investigating. The objective of this research was to evaluate the action of the previously undescribed natural diterpene 3,3′-diisopropyl-2,2′,5,5′-tetramethoxy-6,6′-dimethylbiphenyl-4,4′-diol (biisoespintanolcompound **2**), against several human tumor cell lines (A549, MDA-MB-231, DU145, A2780, A2780-cis) and the non-tumor cell line MRC-5. Experiments with 3-(4,5-dimethylthiazol-2-yl)-2,5-diphenyltetrazolium bromide (MTT) and fluorescence with propidium iodide (PI), 4′,6-diamidino-2-phenylindole dilactate (DAPI), and green plasma revealed the cytotoxicity of **2** against these cells. Furthermore, morphological and chromogenic studies demonstrated the action of **2** on cell morphology and its inhibitory capacity of reproductive viability for colony formation in A549 cells. Furthermore, 3D experiments validated the damage caused by this diterpene in these cells. These results contribute to the search for novel compounds with antitumor potential and serve as a basis for advancing studies into the mechanisms of action of these compounds and the development of synthetic derivatives or analogs with a better antitumor profile.

## 1. Introduction

Cancer is one of the most devastating diseases in the world, considered the second leading cause of death after cardiovascular diseases, and morbidity is expected to increase globally over the next decades [1,2]. This disease affects one in six people worldwide. The disease is more prevalent in individuals with inborn errors of immunity than in immunocompetent individuals, frequently causing death in these people [3,4]. The complexity of this disease is due in part to intratumoral heterogeneity and the dynamic nature of cellular plasticity; the hallmarks of cancer, including metabolism, angiogenesis, metastasis, and the ability to develop resistance, especially to drugs, give it increased survival and proliferative capacity [2,5,6,7], becoming a major challenge in medical practice. Estimates from the *World Health Organization’s* (WHO) *International Agency for Research on Cancer* (IARC) in 2022 suggest that approximately one in five men or women will develop cancer in their lifetime, while approximately one in nine men and one in twelve women will die from it [8], highlighting the growing burden of this disease and the disproportionate impact on underserved populations, considering the urgent need to address cancer inequities at a global level.

The high toxicity and unfavorable side effects of the drugs used in chemotherapy, as well as the resistance to these treatments, have limited their use and made the effective treatment of this disease difficult [9,10,11,12]. Despite technological advances in diagnosis and treatment, cancer remains one of the main threats to humanity. In this context, there is an urgent need to search for and develop novel therapeutic approaches that exhibit antitumor activity while sparing healthy cells. Since ancient times, natural products, particularly those of plant origin, have made a valuable contribution to cancer pharmacotherapy [13,14,15,16], compounds such as polyphenols, flavonoids, terpenoids, and alkaloids activate the antitumor immune response and improve the effectiveness of treatments against this disease [17]; the ability of these compounds to inhibit tumor cell growth both in vitro and in vivo, and their antioxidant, antiproliferative, and proapoptotic effects in various types of cancer have been widely documented [17,18,19,20,21,22,23].

Terpenoids are a chemically diverse group, constituting the largest family of natural products, with members present in almost all life forms and with a wide range of applications [24]. These compounds have been the subject of research in the discovery of antineoplastic drugs, given their potential antitumor effect and low toxicity; in addition, the structural diversity of these compounds provides greater plasticity when interacting with these cells [9,25,26,27]; in particular, phenolic diterpenes have been documented to have a significant cytotoxic effect against different types of cancer [23,28], likewise, their beneficial clinical effects in cancer patients have also been documented [29]. Also, taxane-type diterpenes have been widely used as anti-tumor agents against different types of cancer, highlighting their mechanism of action against microtubules; these compounds can alter microtubule dynamics by inducing mitotic arrest dependent on the mitotic spindle assembly checkpoint; they shift the equilibrium between soluble tubulin and the microtubule polymer in favor of the latter and therefore reduce the critical concentration of tubulin needed for microtubule formation [30,31]. In addition, they alter multiple cellular oncogenic processes, such as mitosis, angiogenesis, apoptosis, inflammatory response, and ROS production, with the consequent cell death [12,30,32]. Likewise, terpenes such as carvacrol, linalool, thymol, D-limonene, citral, β-caryophyllene, and others have demonstrated their antitumor potential through various mechanisms of action that also include alteration of the cell cycle, autophagy, necroptosis, inhibition of cell proliferation, and angiogenesis [33,34,35].

Chemical investigation of *Oxandra xylopioides* (Annonaceae) led to the isolation and structure elucidation of a new phenolic diterpene **2**, along with the identification of two previously known terpenes **1** and **3**. Considering the demonstrated biological potential of some compounds obtained from members of the Annonaceae family, particularly terpenes, we hypothesize that compound **2** has cytotoxic effects against human tumor cell lines. The objective of this study was to evaluate the cytotoxic effect (IC_50_) of the phenolic diterpene **2** against several human cell lines: A549 (lung adenocarcinoma) as a representative and widely used non-small cell lung cancer (NSCLC) model, which is highly relevant given the global impact of lung cancer; MDA-MB-231 (triple-negative breast cancer) provides an aggressive, treatment-resistant phenotype that is useful for evaluating compounds with potential broad antitumor activity; DU145 (androgen-independent prostate cancer) models metastatic, hormone-refractory disease, enabling the assessment of a clinically challenging prostate cancer subtype; A2780 and A2780-cis (ovarian cancer and its cisplatin-resistant variant) enable a direct comparison of sensitive and chemoresistant phenotypes to test whether compound **2** remains active despite platinum resistance and MRC-5 (non-tumor lung fibroblasts) was included as a normal control line to calculate selectivity and evaluate preferential cytotoxicity. Together, this panel represents various tumor origins, genetic backgrounds, and resistance profiles, providing a robust, translationally relevant evaluation of compound **2**. These results point to an interesting path for further investigation into the antitumor mechanisms of action of this natural diterpene, which could, in the future, serve as a template for the synthesis of analogous compounds with a better antitumor effect.

## 2. Results

### 2.1. Isolation and Structure Elucidation of the Compounds

The ethanolic extract of the leaves of *Oxandra xylopioides* was partitioned with petroleum benzine (PB)/H_2_O and fractionated by column chromatography, obtaining the compounds **1**–**3** (Figure 1). The known compounds 3-hydroxy-2,5-dimethoxy-*p*-cymene (**1**) and berenjenol (**3**) were identified by comparing their MS and NMR spectroscopic data with those reported in the literature [35,36].

The new compound **2** was isolated as a crystalline amorphous solid (mp 106–107 °C). Its molecular formula, C_24_H_34_O_6_, implying eight degrees of unsaturation, was deduced using EI-MS and NMR spectra.

The ^1^H-NMR and HMQC NMR spectra of **2** showed the presence of one hydroxylic proton, as well as two methoxys, one methyl, and superimposed signals for one isopropyl group (Table 1). The ^13^C-NMR and DEPT-135 NMR data revealed the further presence of three oxygen-binding aromatic carbons, three substituted aromatic carbons, one carbon from the methine group, four carbons from the methoxy groups, and one carbon from the methyl group. One spin system was assembled using COSY correlations: the isopropyl system 8-H(9-H_3_)/10-H_3_ (Figure 2). These were connected by HMBC correlations, revealing a structure similar to compound **1**. Most significant were the HMBC correlations from H-7 to C-1, C-5, and C-6; from H-8 to C-2, C-3, C-4, C-9, and C-10; and from HO to C-3, C-4, and C-5 in this process. This analysis leads to the conclusion that compound **2** is a biphenyl diterpene formed by the dimerization of compound **1**.

A structurally related compound has been isolated from *Thymus vulgaris* leaves, with activity on human platelet aggregation induced by collagen and thrombin [37] and antioxidative [38]. They have also been isolated from *Anethum sowa* [39], and *Baccharis dracunculifolia* [40].

### 2.2. MTT Assay

Compound **2** showed cytotoxic activity against human tumor cell lines (A549, MDA-MB-231, DU145, A2780, and A2780-cis) and the non-tumor cell line (MRC-5), using the MTT cell survival assay. These results are similar to those reported with compound **1** in the same cell lines [21], with the clear advantage of compound **2** exhibiting approximately 2.5 times lower toxicity in a non-cancerous cell line compared to compound **1** under the same conditions. The IC_50_ values of compounds **1**, **2**, **3,** and cisplatin are shown in Table 2, as well as the selectivity index (SI) of **2** and cisplatin. Cisplatin was used as a positive control. The SI was calculated by dividing the IC_50_ values of the non-tumor cells used as controls by the IC_50_ values of the tumor cell lines. A selectivity index greater than **1** indicates that cytotoxicity on tumor cells exceeds that of control cells. Compound **2** showed slight selectivity against tumor cells (SI > 1) compared to non-tumor cells (MRC5). The results with the triterpene **3** showed no cytotoxic activity at the concentrations tested, even when the compound was used at concentrations above 1000 µM.

### 2.3. Cell Morphology Assay

Compound **2** treatment caused significant morphological changes in lung tumor cells (A549), compared to those observed in the non-tumor cells (MRC-5), in which the changes were less evident (photo in Appendix A). When A549 cells were exposed to the highest concentrations (IC_50_ and 2 IC_50_), several morphological changes were observed: cell condensation and contraction, decreased confluence, reduced cell number, and loss of the original shape. In addition, smaller, spherical cell fragments surrounded by membrane (similar to apoptotic bodies) were evident, some with protuberances in the plasma membrane. The effect on morphology intensified with increasing compound **2** concentration; this progressive and drastic modification of cell shape could suggest death by apoptosis caused by **2** [40,41,42,43,44]. Propidium iodide (PI) staining revealed red-stained cells, indicative of cell death. Propidium iodide (PI) staining revealed red-stained cells, indicative of cell death. Notably, cell death was observed even at the lowest tested concentration (0.5 IC_50_), and the proportion of dead cells increased with rising concentrations, reaching a maximum at IC_50_, so we proposed that compound **2** induced cell death, and this could be through the induction of apoptosis (Figure 3).

### 2.4. Clonogenicity Assay

Lung tumor cells (A549) and the non-tumor cells (MRC-5) were treated with compound **2** (0.5 IC_50_, IC50, and 2 IC_50_) for 48 h to evaluate their survival, reproduction, and colony formation capacity. DMSO solvent was used as a control. We observed the existence of a concentration-response correlation, given the continuous loss of clonogenicity of the cells as the concentration of compound **2** increased, demonstrating the cytotoxic and cytostatic effect of this diterpene on A549 tumor cells. The cells were evaluated after a period of 10 days. The number of colonies of A549 cells treated with compound **2** decreased significantly compared to the number of colonies of MRC-5 cells, where only at the highest concentration (2 IC_50_) showed a slight decrease in the number of colonies formed (Figure 4). The antiproliferative effect of compound **2** was substantially greater in A549 cells than in MRC-5 cells, demonstrating the effect of compound **2** against the reproductive viability of these cells and that this is a dose-dependent effect.

### 2.5. Three-Dimensional Assays

The study in three-dimensional cell culture models is of great relevance since it allows us to accurately imitate the tumor microenvironment in vivo; cells cultured in these environments retain their natural three-dimensional physical form, generally in the form of spheroids, which facilitates regulatory mechanisms and signaling networks between cells. In addition, we can test the efficacy of drugs in an in vitro model comparable to in vivo tumors [43,45]. Analysis of the compound **2** results in spheroids from the lung tumor cells (A549) shows a reduction in spheroid size at a concentration of 125 µM, after 144 h of treatment. Upon increasing the concentration to 250 µM, the spheroid radius increased, likely due to reduced intercellular interactions and loss of spheroid integrity (Figure 5). DAPI and PI staining (Figure 6) clearly demonstrates cell death, with a dose-response relationship even at the lowest concentrations tested, with an increase in the number of dead cells as the concentration increased. At 250 µM, spheroid disorganization and increased cell spacing were even more evident (Figure 5).

## 3. Discussion

Cancer remains one of the leading causes of mortality worldwide, and lung cancer, in particular, is responsible for the highest number of deaths among all cancer types [8]. Despite significant advances in molecularly targeted therapies and early detection technologies, resistance, toxicity, and relapse continue to limit the success of current chemotherapeutic strategies [46,47]. In this context, natural products and their derivatives represent a valuable source of structurally diverse molecules capable of interfering with key cellular processes associated with cancer progression [14,17,28]. Besides phenolic compounds, alkaloids, flavonoids, triterpenoids, and diterpenes have demonstrated broad cytotoxic and chemopreventive potential through mechanisms involving apoptosis, modulation of oxidative stress, and interference with proliferation and angiogenesis [14,48,49].

In the present study, the phenolic diterpene compound **2** isolated from *Oxandra xylopioides* showed measurable cytotoxic activity against a panel of tumor cell lines, including A549 (lung), MDA-MB-231 (breast), DU145 (prostate), and A2780/A2780-cis (ovary), as well as moderate selectivity over non-tumor MRC-5 cells. The IC50 values (51.9–73.3 µM in tumor cells) confirm a reproducible dose- and time-dependent antiproliferative effect. These values are within the range typically reported for phenolic monoterpenes and diterpenes with confirmed antitumor activity, such as thymol and carnosic acid, respectively [28,50]. Although the selectivity indices (SI = 1.3–1.9) indicate only a modest tumor preference, they nonetheless suggest a therapeutic window that could be further optimized by structural modifications or formulation approaches.

The biological effects of compound **2** were consistent across several cellular assays. Morphological analysis of A549 lung tumor cells revealed apoptotic features, including cell shrinkage, nuclear condensation, reduced confluence, and the formation of membrane-bound cell fragments, often resembling apoptotic bodies. These alterations, which were intensified at concentrations close to or higher than the IC_50_, are consistent with those induced by other phenolic terpenes, such as thymol and isoespintanol [21,50]. Propidium iodide staining confirmed the loss of membrane integrity, while nuclear fragmentation observed under DAPI fluorescence supports an apoptosis-like cell death process [51,52]. Considering the hydrophobic and biphenolic nature of compound **2**, its lipophilicity probably facilitates the interaction with cell membranes, increasing permeability and compromising membrane potential, a mechanism frequently reported for phenolic terpenes [10,42,50,51,52,53,54].

The clonogenic assay further demonstrated that compound **2** exerts a pronounced cytostatic effect by reducing colony formation in A549 cells in a concentration-dependent manner. The inhibition of reproductive viability at concentrations of IC_50_ and 2IC_50_, in contrast to the minor effect on non-tumor MRC-5 cells, highlights its preferential interference with tumor proliferation. The ability to affect clonogenic survival is biologically relevant, as it correlates with suppression of tumor regrowth and metastatic potential [55,56,57]. Thus, beyond acute cytotoxicity, compound **2** appears to affect long-term proliferative capacity, a valuable feature in potential anticancer agents.

The results obtained from three-dimensional spheroid models reinforce the cytotoxicity observed in monolayer cultures. In A549 spheroids, compound **2** induced progressive disorganization, reduction in spheroid integrity, and increased PI staining in a dose-dependent manner. The increase in spheroid radius at higher concentrations (250 µM) is likely associated with structural collapse rather than cell proliferation, reflecting decreased intercellular cohesion and spheroid disaggregation. The maintenance of cytotoxic activity under three-dimensional conditions where nutrient gradients, hypoxia, and intercellular interactions better mimic tumors in vivo suggests that compound **2** retains its efficacy in a more physiologically relevant microenvironment.

Together, these findings suggest that compound **2** exerts its antitumor effects primarily through cytostatic and proapoptotic mechanisms, consistent with other phenolic diterpenes that disrupt redox homeostasis, mitochondrial function, and proliferative signaling [28,32,33,58]. Its moderate selectivity and consistent activity across different tumor models point to a generalizable mechanism of action, rather than a cell line-specific effect. The combination of two-dimensional, clonogenic, and three-dimensional data provides converging evidence that this compound interferes with key pathways of tumor cell survival and spread.

In summary, biisoespintanol (compound **2**) appears to be a promising phenolic diterpene with antitumor potential, characterized by dose-dependent cytotoxicity, morphological features of apoptosis, inhibition of clonogenic proliferation, and maintained activity in 3D spheroid models. These results support the continued investigation of this molecule as a backbone for diterpene-based anticancer agents and suggest its potential as an adjuvant candidate to conventional therapies. Future studies should focus on elucidating their molecular targets and refining their structure-activity relationships to improve selectivity and potency, while maintaining their advantages as a natural scaffold.

## 4. Materials and Methods

### 4.1. General Chemical Procedures

IR spectra were recorded with a Thermo Scientific FT-IR spectrophotometer, NICOLET IS5 (Madison, WI, USA). EIMS were obtained with a Nermag-Sidar R10-10C mass spectrometer (Argenteuil, France). Then, 1D and 2D nuclear magnetic resonance (NMR) spectra were obtained using a Bruker Avance DRX 400 MHz spectrometer (Madison, WI, USA), ^1^H 400 MHz, ^13^C 100 MHz. Column chromatography was performed with silica gel 60 (0.063–0.2 mm), and silica gel 60 F_254_ chromatography plates (Merck 60 F254 0.2 mm) were used. All chemicals and solvents were acquired from Merck KGaA (Darmstadt, Germany) in analytical grade.

### 4.2. Plant Material and Obtaining the Extracts

The leaves of *Oxandra xylopioides* were collected in October 2019 from a specimen located in the Municipality of Monteria, Department of Córdoba, with coordinates 08°48′17″ north latitude and 75°42′07″ west longitude. A herbarium specimen is deposited in the Joaquin Antonio Uribe Botanical Garden of Medellin, Colombia, with the collection number JAUM 037849. The plant material, free of impurities, was dried at room temperature and pulverized in a knife mill. The dried and ground material (4400 g) was extracted by percolation with 96% ethanol until exhaustion. The extract was concentrated in a rotary evaporator (Hei-VAP Core, Wood Dale, IL, USA) until 490 g of ethanolic extract.

### 4.3. Isolation of the Compounds

Initially, 50 g of ethanolic extract was partitioned with petroleum benzine 40–60 °C (PB)/H_2_O to obtain 17 g of PB extract. Subsequently, 10 g of PB extract was fractionated by column chromatography (silica gel 350 g, PB/dichloromethane (DCM) 2:1, to PB/DCM/MeOH 2:3:1), obtaining 38 fractions. Fraction F5 (3491 mg) was recrystallized from hexane, obtaining 2830 mg of compound **1**. Fraction F19 (453 mg) was recrystallized from hexane, obtaining 252 mg of compound **2**. Fraction F25 (197 mg) was recrystallized in methanol, obtaining 45 mg of compound **3**.

### 4.4. Spectral Data

Compound **1**: 2,5-dimethoxy-3-hydroxy-*p*-cymene (isoespintanol): crystalline amorphous solid; the EI-MS: [M]^+^ *m*/*z* 210 (49%) and fragments *m*/*z* 195 (100%), 180, 165, 150, 135, and 91. ^1^H-NMR (400 MHz, CDCl_3_): δ 6.22 s, 1H (H-6), δ 5.85 s, 1H (HO-3), δ 3.77 s, 3H (H-12), δ 3.76 s, 3H (H-11), δ 3.52 hep, *J* = 7.1 Hz, 1H (H-8), δ 2.29 s, 3H (H-7), δ 1.33 d, *J* = 7.1 Hz, 6H (H-9, and H-10). ^13^C-NMR (100 MHz, CDCl_3_): δ 154.3 (C-5), δ 147.4 (C-3), δ 139.7 (C-2), δ 126.8 (C-1), δ 120.4 (C-4), δ 104.4 (C-6), δ 24.6 (C-8), δ 60.8 (C-11), δ 55.7 (C-12), δ 20.6 (C-9, C-10), δ 15.8 (C-7). Data are in good agreement with those of [36].

Compound **2**: 3,3′-diisopropyl-2,2′,5,5′-tetramethoxy-6,6′-dimethylbiphenyl-4,4′-diol (biisoespintanol): crystalline amorphous solid (mp 106–107 °C); IR, 3454 cm^−1^, 2958 cm^−1^, 2933 cm^−1^, 1452 cm^−1^, 1409 cm^−1^, 1100 cm^−1^, 1016 cm^−1^. The EI-MS: [M]^+^ *m*/*z* 418 (100%) and fragments *m*/*z* 403, 388, 387, 346, 43. ^1^H-NMR (400 MHz, CDCl_3_): δ 5.80 s, 2H (HO-4, and HO-4′), δ 3.78 s, 6H (H-11, and H-11′), δ 3.31 s, 6H (H-12, and H-12′), δ 3.44 hep, *J* = 7.0 Hz, 2H (H-8, and H-8′), δ 1.93 s, 6H (H-7, and H-7′), δ 1.39 d, *J* = 7.0 Hz, 6H (H-10, and H-10′), δ 1.38 d, *J* = 7.0 Hz, 6H (H-9, and H-9′). ^13^C-NMR (100 MHz, CDCl_3_): δ 152.4 (C-2, and C-2′), δ 147.3 (C-4, and C-4′), δ 142.2 (C-5, and C-5′), δ 127.0 (C-6, and C-6′), δ 125.3 (C-3, and C-3′), δ 123.1 (C-1, and C-1′), δ 25.9 (C-8, and C-8′), δ 60.8 (C-11, and C-11′), δ 60.7 (C-12, and C-12′), δ 21.0 (C-9 and C-9′), δ 21.1 (C-10 and C-10′), δ 13.2 (C-7, and C-7′).

Compound **3**: Berenjenol: Colorless amorphous solid; EIMS *m*/*z* 456 (15%), 441, 438, 423 (37), 273 (100%). The 1D and 2D NMR data (^1^H 400 MHz, ^13^C 100 MHz) are in good agreement with those of [59].

### 4.5. Reagents

For cytotoxicity assays, Dulbecco’s modified Eagle’s medium (DMEM) and RPMI 1640, gentamicin, amphotericin B, dimethyl sulfoxide (DMSO), trypan blue, and 3-(4,5-dimethylthiazol-2-yl)-2,5-diphenyltetrazolium bromide [MTT] were obtained from Sigma-Aldrich^®^ (St. Louis, MO, USA). Crystal violet (CV) and phosphate-buffered saline (PBS) were obtained from Synth (Diadema, São Paulo, Brazil). Propidium iodide was obtained from BD Biosciences (San Jose, CA, USA). Trypsin, fetal bovine serum (FBS), and penicillin were obtained from Vitrocell (Campinas, São Paulo, Brazil). Sodium bicarbonate was obtained from Neon (Suzano, São Paulo, Brazil). Plasma green, CellMask™, and DAPI (4′,6-diamidino-2-phenylindole dilactate) were obtained from Invitrogen by Thermo Fisher Scientific (Eugene, OR, USA).

### 4.6. Cell Lines

The cell lines A549 (human lung alveolar basal cell epithelial adenocarcinoma, ATCC No. CCL-185), MDA-MB-231 (human breast triple-negative adenocarcinoma of mesenchymal phenotype, ATCC No. HTB-26), DU145 (prostate tumor, ATCC No. HTB-81), A2780-cis (cisplatin-resistant human ovarian tumor, ECACC No. 93112517), A2780 (human ovarian tumor, ECACC No. 93112519), and MRC-5 (human non-tumor lung, ATCC No. CCL-171) were obtained from the American Type Culture Collection (ATCC) (Rockville, MD, USA).

### 4.7. Cell Cultures

The A549, MDA-MB-231, DU145, and MRC-5 cell lines were cultured in DMEM, while the A2780-cis and A2780 cell lines were cultured in RPMI 1640 medium, both supplemented with 10% fetal bovine serum (FBS) and 1% penicillin G sodium salt/streptomycin sulfate, and maintained in a humidified atmosphere with 5% CO_2_ at 37 °C. Subcultures were performed twice a week until approximately 80% confluence was reached, using a Nikon Eclipse TS 100 inverted microscope (Nikon, Melville, NY, USA) for the assays.

### 4.8. MTT Assay

The MTT reagent was used to evaluate the cytotoxic effect of **2** (IC_50_: 50% inhibitory concentration of the cell population) against the cell lines in this study, following the methodology described in [21]. This assay determines the ability of metabolically viable cells to reduce the tetrazolium rings of the reagent and form formazan crystals; therefore, the number of viable cells is directly proportional to the level of formazan produced [60,61,62]. Cisplatin was used as the reference compound [11,25]; therefore, the commercial cisplatin-resistant cell line A2780-cis was used. For the assays, cells (1.5 × 10^4^ cells/well) were seeded in 150 µL of RPMI 1640 (for A2780 and A2780-cis) and DMEM (for A549, MDA-MB-231, DU145, and MRC-5) using 96-well microplates (Kasvi, São José dos Pinhais, Brazil). For adhesion and proliferation, the microplates were incubated at 37 °C for 24 h with 5% CO_2_. Subsequently, 0.75 µL of the compound **2** stock solution dissolved in DMSO was added to reach final concentrations of 15.62, 31.25, 62.5, 125, and 250 µM in the final reaction wells. The microplates were incubated at 37 °C for 48 h. MTT (50 µL, 1 mg/mL) was then added to each well. The plates were incubated with 5% CO_2_ at 37 °C for 4 h. The MTT was then discarded, and the microplates were dried at room temperature; the formazan crystals produced were then dissolved in 100 µL of DMSO. Finally, absorbance readings (OD 540 nm) were taken using an Epoch 2 microplate reader (Biotek, Winooski, VT, USA). DMSO-treated and untreated cells were used as controls. Experiments were performed three times in triplicate. Likewise, cytotoxicity experiments were performed under the same conditions with the natural triterpene berenjenol **3** obtained from *O. xylopioides*. The test results were expressed as dose–response curves using the concentrations described previously. IC_50_ values were calculated from the fit (R^2^ > 0.95) of the Hill slope curve of the experimental data using nonlinear regression analysis in GraphPad Prism version 8.0.1 software.

### 4.9. Cell Morphology Assay

The effect of compound **2** on cell morphology was determined using the A549 cell line as a model. For the assay, 1500 µL of a cell suspension (1 × 10^5^ cells/well) was seeded in 12-well plates and incubated at 37 °C for 24 h and 5% CO_2_. Subsequently, the cells were treated with 7.5 µL of **2** (0.5 IC_50_, IC_50_, and 2 IC_50_) and incubated at 37 °C for 48 h and 5% CO_2_. Then, the cells were observed and photographed at 0 h and 24 h. The culture medium was then removed, and the cells were fixed with 1 mL of methanol for 10 min. The methanol was then discarded, 500 µL of PBS and 200 µL of green plasma were added to the wells, and the microplates were incubated for 30 min. Subsequently, the green plasma was removed, 500 µL of PBS and 200 µL of DAPI were added, and the plates were incubated again for 5 min. PI staining (100 µL) was also performed for 10 min. Cells were then photographed using the CELENA^®^ S Digital Imaging System (Logos Biosystems, Annandale, VA, USA). The morphology of cells treated with **2** was compared with that of cells treated with 0.5% DMSO, used as a negative control. Experiments were performed in triplicate.

### 4.10. Clonogenicity Assay

The A549 cell line was also used to evaluate the reproductive viability of cells after compound **2** treatment using the clonogenicity assay [21,63]. The criterion used was the number of colonies. In the assay, 1500 µL of the cell suspension (1000 cells/well) were seeded in 6-well plates and incubated at 37 °C with 5% CO_2_ for 24 h. Cells were then treated with 10 µL of 2 (0.5 IC_50_, IC_50_, and 2 IC_50_) and incubated for 48 h. The culture medium was then discarded from the plates, cells were washed with 2 mL of PBS and 2 mL of DMEM culture medium supplemented with 10% FBS, and the plates were incubated (5% CO_2_ at 37 °C) for 10 days. Finally, the supernatant was discarded, and the cells were washed with PBS and fixed with a solution of methanol (2 mL) and acetic acid (3:1 (*v*/*v*)) for 5 min. The colonies were stained with crystal violet (0.05% crystal violet, 1% formaldehyde, 1× PBS, and 1% methanol). Colony counting was performed using ImageJ 1.53t (Wayne Rasband, NIH, Bethesda, MD, USA), Java 1.8.0_345 image analysis software [58], and Prisma version 8.0 software. To determine the plating efficiency [PE] (which determines the number of colonies formed based on the number of cells plated) and the survival fraction [SF] (which determines the number of colonies developed after applying **2** treatments) [64], the following formulas were used [65]:PE = Number of colonies formed/number of cells seeded × 100%(1)SF = Number of colonies formed after treatment/number of cells seeded × PE(2)

### 4.11. Three-Dimensional Assays

For the 3D cell culture assay, the 96-well Bioprint kit from Magnetic 3D Cell Culture Technology (m3D-Greiner) was used. First, 100 µL of a nanoparticle-containing solution (NanoShuttle—PL) was added to a cell culture flask containing the A549 cell line. After 24 h, the nanoparticles containing cells were seeded in a 96-well repellent culture plate (1500 cells/well). The culture dish was placed under a magnetic impeller to form spheroids. The dish was kept in an incubator (37 °C, 5% CO_2_), and spheroid formation and growth were monitored using a CELENA^®^ S Digital Imaging System (Logos Biosystems). After 48 h, different concentrations of the compound were added to the spheroids, and the experiment was monitored for 144 h. The fluorescent markers DAPI and PI were added on the last day of treatment.

### 4.12. Data Analysis

The results were analyzed using GraphPad Prism version 8.0.1 software (GraphPad Software/Dotmatics) and ImageJ 1.53t image analysis software (Wayne Rasband, NIH, Bethesda, MD, USA), Java 1.8.0_345.5.

## 5. Conclusions

The phenolic diterpene **2** has cytotoxic activity against the tumor cell lines A549, MDA-MB-231, DU145, A2780-cis, A2780, and the normal lung cell line MRC-5. The results show that **2** has an antiproliferative effect against these cells, affecting cell morphology and interrupting their reproductive viability. Therefore, **2** becomes a compound of interest to continue investigating the mechanisms of action against tumor cells. Furthermore, it could be used as a natural template for the synthesis or semi-synthesis of analogs, thus contributing to the search for alternatives that facilitate the control of complex diseases such as cancer.

## Figures and Tables

**Figure 1 molecules-30-04629-f001:**
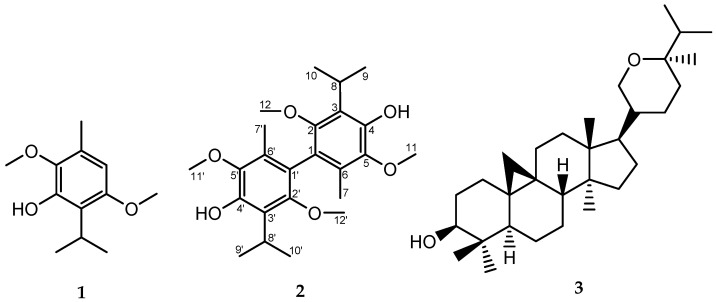
Chemical structures of compounds isolated from the leaves of *Oxandra xylopioides*. **1**, isoespintanol; **2**, biisoespintanol; and **3**, berenjenol.

**Figure 2 molecules-30-04629-f002:**
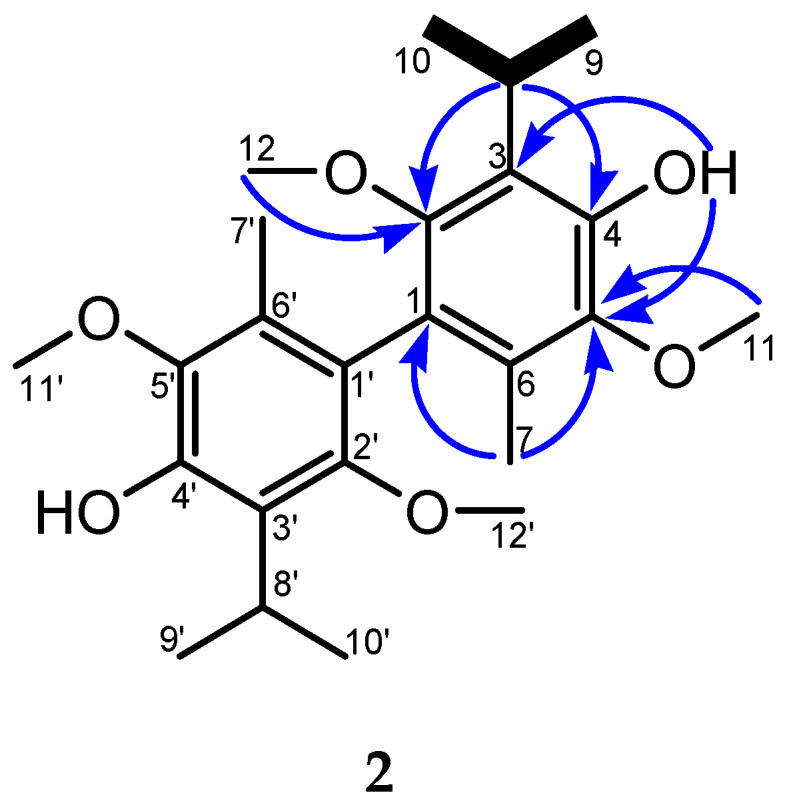
Key COSY (bold bonds) and HMBC (arrows) some correlations for **2**.

**Figure 3 molecules-30-04629-f003:**
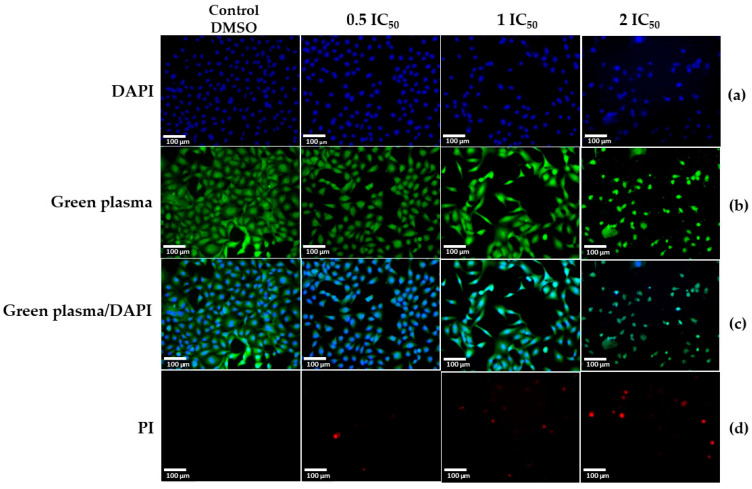
Morphological changes in lung tumor cells (A549) after treatment with compound **2** (0.5 IC_50_, IC_50_, and 2 IC_50_) at 48 h. Fluorescence microscopy with plasma green/DAPI (**a**–**c**) and propidium iodide (**d**); 10× magnification. Significant morphological changes were observed with the IC_50_ and 2 IC_50_ treatments at 48 h. Cells were photographed using the CELENA^®^ S digital imaging system (Logos Biosystems, Anyang-si, Republic of Korea) and are shown at a scale of 100 µm.

**Figure 4 molecules-30-04629-f004:**
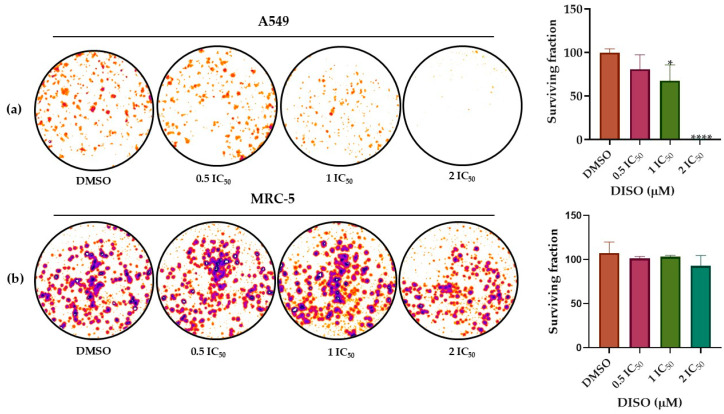
Colony formation in lung tumor (A549) and non-tumor (MRC-5) cell lines following treatment with compound **2** (0.5 IC_50_, IC_50_, and 2 IC_50_) after 10 days of incubation; 1.1× increase. In (**a**), the results of Dunnett’s test with a 95% confidence level showed values of * *p* < 0.05 (DMSO vs. IC_50_) and **** *p* < 0.0001 (DMSO vs. 2 IC_50_), which indicates that there are statistically significant differences between the number of colonies formed in A549 cells treated with different concentrations of **2** and the results shown with the control group (DMSO). In (**b**), MRC-5 cells, the results of the Dunnett test with a confidence level of 95% indicate that there are no statistically significant differences between the treatment with DMSO and the different concentrations of **2** evaluated in the colony formation of these cells.

**Figure 5 molecules-30-04629-f005:**
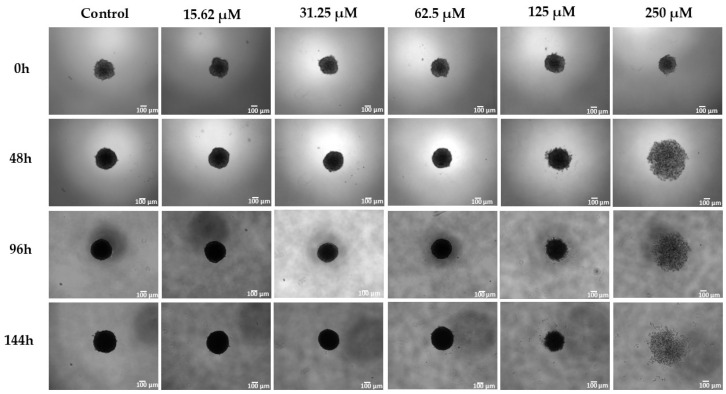
Brightfield microscopy of A549 cell spheroids treated with compound **2** (15.62, 31.25, 62.5, 125, and 250 µM) and untreated (control); 4× magnification. Cell detachment was evident after 8 days of **2** treatments, and is shown at a scale of 100 µm.

**Figure 6 molecules-30-04629-f006:**
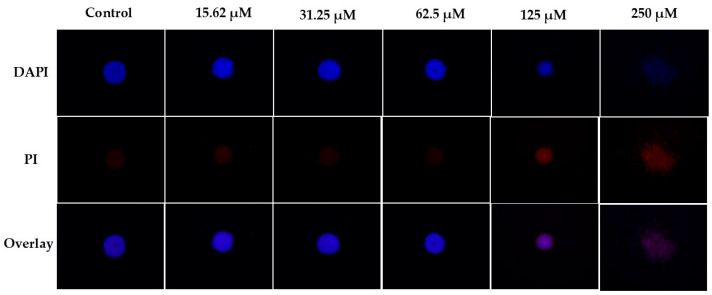
Fluorescence microscopy of lung tumor cells (A549) spheroids, labeled with DAPI and PI, treated with compound **2** (15.62, 31.25, 62.5, 125, and 250 µM) and untreated, at 4× magnification. PI staining shows cell death at all concentrations.

**Table 1 molecules-30-04629-t001:** NMR spectroscopic data (^1^H 400 MHz, ^13^C 100 MHz) of **1** in CDCl_3_.

Pos.	δ_C_, Type	Δ_H_ Mult. (*J* in Hz)	COSY	HMBC
1/1′	123.1, C			
2/2′	152.4, C			
3/3′	125.3, C			
4/4′	147.3, C			
5/5′	142.2, C			
6/6′	127.0, C			
7/7′	13.2, CH_3_	1.93, s		1, 5, 6/1′, 5′, 6′
8/8′	25.9, CH	3.44, hep (7.0)	9, 10/9′, 10′	2, 3, 4, 9, 10/2′, 3′, 4′, 9′, 10′
9/9′	21.0, CH_3_	1.38, d (7.0)	8/8′	3, 8, 10/3′, 8′, 10′
10/10′	21.1, CH_3_	1.39, d (7.0)	8/8′	3, 8, 9/3′, 8′, 9′
11/11′	3.78, CH_3_	3.78, s		5/5′
12/12′	3.31, CH_3_	3.31, s		2/2′
4-OH/4′-OH		5.80, s		3, 4, 5/3′, 4′, 5′

**Table 2 molecules-30-04629-t002:** IC_50_ values (µM) of compounds **1**, **2**, **3,** and cisplatin against monolayers of the tumor cell lines A549, MDA-MB-231, DU145, A2780, A2780-cis, and the non-tumor cell line MRC-5 and selectivity index (SI), after 48 h of treatment.

Cell Lines	1 **	2	SI	Cisplatin	SI	3 (IC_50_)
A549	59.7 ± 2.74	63.70 ± 1.61	1.54	14.40 ± 1.40	0.86	>1000
MDA-MB-231	52.39 ± 3.20	73.32 ± 1.60	1.34	2.40 ± 0.21	5.19	>1000
DU-145	47.84 ± 3.52	51.95 ± 4.93	1.89	2.30 ± 0.42	5.42	>1000
A2780cis	60.35 ± 8.40	55.25 ± 2.03	1.78	25.61 ± 0.29	0.48	>1000
A2780	42.15 ± 1.39	68.28 ± 2.97	1.44	11.17 ± 0.30	1.11	>1000
MRC5 *	39.95 ± 3.76	98.63 ± 0.98	-	12.47 ± 0.15	-	>1000

* The non-tumor line MRC5 was used to calculate the SI, which was calculated by the ratio of IC_50_ (non-tumor cells)/IC_50_ (tumor cell lines). ** Data obtained from reference [21].

## Data Availability

The data presented in this study are available in the article.

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
