# Peer review of "A New Diterpene with Cytotoxic Potential Against Human Tumor Cells"

_molecules, 2025, doi:10.3390/molecules30234629_

Round 1
Reviewer 1 Report
Comments and Suggestions for Authors
Paper entitled: “New diterpene with cytotoxic potential against human tumor 2 cells ” (molecules-3979953). The subject of the study and the obtained results seem very interesting; however, there are some issues that should be addressed before publication in Molecules.
L73: Please clarify how diterpenes interfere with microtubule dynamics (e.g., inhibition of polymerization, stabilization/destabilization, binding site if known) and provide appropriate literature citations to support the statement. Including one or two recent, primary references that describe the molecular mechanism would strengthen this claim.”
L86: It is unclear whether the reported cytotoxic activity applies to all tested human tumor cell lines or only to a subset. Please specify explicitly whether cytotoxicity was observed across all human tumor cell lines used in this study, and if not, indicate which lines were responsive versus non-responsive.
L88: The Aim of the study should justify the selection of the particular human tumor cell lines used. Please explain why these specific cell lines were chosen (e.g., relevance to a particular cancer type, representative genetic backgrounds, prior sensitivity/resistance to microtubule-targeting agents, availability, or relevance to the study hypothesis). A short rationale will help readers understand the translational or biological relevance of the chosen models.
Figure 1: Please clarify what labels 1, 2, and 3 represent in the figure description. The current caption does not explain the meaning of these numbers, making interpretation difficult.
Were the observed morphological changes in the cells quantitatively assessed? If so, please provide the methodology and results. If not, please comment on why quantitative analysis was not performed.
Figure 3: The scale bars shown in the images are not clearly visible, even when the figure is enlarged. Please improve the visibility and contrast of the scale bars.
In addition, please include the scale information directly in the figure caption for clarity.
The caption mentions morphological changes after 24 h and 48 h, but these time points are not indicated in the figure itself. Please add these labels to the figure panels.
Figure 4: Please clearly label which dish in the schematic corresponds to each concentration of the tested compound. Without this information, the figure cannot be interpreted accurately.
Figure 5: The images in Figure 5 appear to lack scale bars. Please add appropriate scale bars to all panels.
L227: wording “across multiple cellular assays“ - the phrase ‘across multiple cellular assays’ appears overstated in this context. I recommend using a more moderate expression, such as ‘across several cellular assays’ to reflect the scope of the experiments more accurately.”
L234: A literature reference is needed to support this statement. Please add an appropriate citation.”
In the Data Analysis section, please include full information about the GraphPad Prism software used, including the company name and location of the manufacturer.
Reviewer 2 Report
Comments and Suggestions for Authors
This manuscript reports the isolation and structural elucidation of a new phenolic diterpenoid from the leaves of Oxandra xylopiodes. It also describes the effects of this new compound on several cultured tumor cell lines.
1) The manuscript should be carefully proofread by a native English speaker, preferably one with expertise in natural products chemistry, as there are several issues with the language. For example, on lines 42-23, the sentence “in people with inborn errors of immunity, this disease is more prevalent than in immunocompetent individuals” should be revised to “the disease is more prevalent in individuals with inborn errors of immunity than in immunocompetent individuals”.
2) The abstract contains several logical and editorial problems that reduce clarity and undermine the argument. For example, in the sentence “In this context, natural products, particularly those obtained from plants, play an important role as a source of specialized metabolites with recognized pharmacological properties against cancer, making them an excellent alternative to be investigated.”, If plant-derived metabolites already have “recognized pharmacological properties”, it is unclear why they are described as needing investigation. Please review the whole document carefully and correct it.
3) The HMBC correlations described on lines 114–115 do not match those shown in Figure 2. Furthermore, these correlations do not provide evidence for the connection between C-1 and C-1’.
4) The authors propose that the new compound could serve as a potential lead for an anti-cancer agent. However, its cytotoxic activity is weaker than that of the positive control, cisplatin.
5) Additional evidence is needed to support the apoptosis-inducing activity of the new compound in A549 cells. For example, its effects on caspase-3 activation and PARP cleavage should be examined.
6) On line 55, the phrase “the resistance expressed toward these treatments” should be revised to the more natural wording “the resistance to these treatments.”
7) On lines 58-59, the sentence “In this context, the search and development of novel therapeutic approaches that are antitumor, with minimal effects on healthy cells is urgent.” is difficult to understand. It should be revised to “In this context, there is an urgent need to search for and develop novel therapeutic approaches that exhibit antitumor activity while sparing healthy cells.”
8) On lines 84–86, the authors state: “Considering the biological potential demonstrated in members of the Annonaceae family, particularly terpenes, we hypothesize that 2 has cytotoxic effects against human tumor cell lines.” However, the rationale for assuming that 2 shows strong activity appears weak. The connection between the reported activities of Annonaceae-derived compounds and the expected activity of 2 is not clearly explained.
9) On lines 90–93, the authors state: “These promising results contribute to the search for compounds obtained from natural sources with cytotoxic potential against tumor cells and are a basis for future studies to elucidate the mechanisms of action of these compounds.” However, this conclusion appears to be overstated. The results may not be sufficient to support such broad claims.
10) On lines 134-135, the meaning of the sentence “The results with the triterpene 3 showed no cytotoxic activity at the concentrations tested, only when the compound was used at concentrations above 1000 μM. is not clear. Do the authors mean to say: “The results with triterpene 3 showed no cytotoxic activity at the concentrations tested, even when the compound was used at concentrations above 1000 μM.”?
11) On lines 144-146, the meaning of the sentence “When A549 cells were exposed to the highest concentrations (IC50 and 2 IC50) cell condensation and contraction, decreased confluence, decreased cell number, loss of the original shape was observed.” is not clear. Do the authors mean to say: “When A549 cells were exposed to the highest concentrations (IC50 and 2×IC50), several morphological changes were observed: cell condensation and contraction, decreased confluence, reduced cell number, and loss of the original shape.”?
12) On lines 150-153, the meaning of the sentence “Propidium iodide (PI) staining showed cells stained red, evidencing cell death, even when the cells were treated with the lowest concentration of 0.5 IC50, increasing this behavior as the IC50 values increased,” is not clear. Do the authors mean to say: “Propidium iodide (PI) staining revealed red-stained cells, indicative of cell death. Notably, cell death was observed even at the lowest tested concentration (0.5 × IC50), and the proportion of dead cells increased with rising concentrations, reaching a maximum at IC50.”?
13) On lines 213–216: The biological activities cited by the authors—apoptosis induction, modulation of oxidative stress, and inhibition of proliferation and angiogenesis—are not unique to diterpenoids. These mechanisms are broadly shared among various classes of natural products, including flavonoids, alkaloids, triterpenoids, and others. Therefore, the statement as written overemphasizes the distinctiveness of diterpenes and could be misleading unless supported by specific comparative data. The authors may consider revising the sentence to avoid implying exclusivity or to provide evidence demonstrating why diterpenes should be highlighted over other metabolite classes.
14) The title of the manuscript should be revised to “A new diterpene with cytotoxic potential against human tumor cells”.
15) On line 31, “monoterpene” should be revised to “diterpene”.
16) On line 41, “[1,2];” should be revised to “[1,2].”.
17) On line 84, “known terpenes 1,3.” should be revised to “known terpenes 1 and 3.”.
18) Compound numbers should be presented in bold font throughout the manuscript.
19) On line 98, the chemical name of 1 should be revised to 3-hydroxy-2,5-dimethoxy-p-cymene.
20) On line 99, “Berenjenol” should be revised to “berenjenol”.
21) In Figure 1, please show the stereochemistry at C-5 and C-8 in the structure drawing of 3.
22) On line 107, “methoxyls” should be revised to “methoxys”.
23) On line 223, thymol is not a phenolic diterpenoid; it is a phenolic monoterpenoid.
24) Please provide the melting point of 2 obtained as a crystalline–amorphous solid.
25) On line 342, “penicillin/streptomycin” should be revised to “penicillin G Sodium salt/streptomycin sulfate”.
26) On line 348, please remove the word “evaluated”.
27) For the incubation conditions, please describe the temperature first, followed by the time throughout the manuscript.
Comments on the Quality of English LanguageThe manuscript should be carefully proofread by a native English speaker, preferably one with expertise in natural products chemistry, as there are several issues with the language.
Reviewer 3 Report
Comments and Suggestions for Authors
The work of Contreras-Martínez et al. describes the isolation and preliminary biological activity studies of a novel terpene biisoespintanol (compound 2), a dimeric form of isoespintanol (compound 1), which was extensively studied previously by the same group. Compound 2 showed moderate anti-proliferative activities in common human cancer cell lines (Table 2), close to those observed previously for 1 (see Ref. 21). The only obvious advantage of 2 was its ~2.5-fold lower toxicity to a non-cancer line compared with 1 under the same conditions. This fact should be clearly mentioned in the work, and the IC50 values for 2 in Table 2 should be shown alongside the published values for 1 in the same cell lines (Ref. 21). There is no obvious sense in comparing the activities of 2 with those of a structurally unrelated compound, 3. Other suggested changes are as follows:
- The first three sentences of the Abstract (lines 16-22) are too general and should be removed. It is also advisable to cut the general discussion of cancer and anticancer drugs in the Introduction (lines 39-65).
- Systematic and common names of compound 2 should be given in the Abstract, rather than referring to it by the number only.
- Throughout the text, the compound numbers (1-3) should be in the bold font (e.g., lines 98, 99, 103, 106, 113, 155, 116, 217, 227, 235, 239, 245, 256, 264, 366, 372).
- The title of Table 2 should include the treatment time (48 h). The treatment time should also be mentioned in the main text next to the treatment concentrations or IC50 values, e.g., in lines 191 and 221.
- The values in Table 2 should use dots, not commas, to designate decimal places (see the error values in the numbers for 2). The numbers of significant figures should match the error values (e.g., 64 ± 2, not 63.70 ± 1.61). The meaning of SI values in Table 2 is different for 2 and cisplatin. For the former, it is the IC50 value in non-cancer MRC5 cells divided by the value in a cancer cell line. For the latter, it is the IC50 value for 2 divided by the value for cisplatin in the same cell line. This difference should be either explained or changed.
- The caption of Figure 3 mentions that the data are presented for three time points (0, 24 and 48 h), while the figure itself apparently shows only one time point.
- The mention of petroleum benzine in line 292 should include its boiling point.
- The list of references should include either abbreviated or non-abbreviated journal names, but not a mixture of both.
- For the Supplementary Material, it is advisable to show the results of chemical characterization (Figures S3-S8) first, followed by the results of cell assays (Figures S1-S2).
- For the mass spectrometry data (Figure S4), a full-scale spectrum (e.g., m/z = 0-1000) should be presented, along with a zoomed view of the target signal and its calculated isotopic distribution. The figure caption should mention the concentration of the compound and the solvent used.
Round 2
Reviewer 1 Report
Comments and Suggestions for Authors
In the supplementary material, the scale bars in Figures S7 and S8 are completely indistinguishable.
It is also barely visible in Figure 3 of the manuscript
Comments on the Quality of English LanguageImproved
Author Response
|
Response to Reviewer 1 Comments
|
||
|
1. Summary |
|
|
|
Thank you very much for taking the time to review this manuscript. Please find the detailed responses below and the corresponding revisions/corrections highlighted/in track changes in the re-submitted files. |
||
|
2. Questions for General Evaluation |
Reviewer’s Evaluation |
Response and Revisions |
|
Does the introduction provide sufficient background and include all relevant references? |
Can be improved
|
We appreciate your feedback. We have taken your suggestions into account and made the corresponding corrections shown below. |
|
Is the research design appropriate? |
Can be improved |
We appreciate your kind suggestion. The design is appropriate and supported by studies by other researchers. It is certainly susceptible to improvement and that is why we are committed to improving each procedure we develop on a daily basis. |
|
Are the methods adequately described? |
Can be improved |
We appreciate your comment. The methods are sufficiently described; however, we are now attaching complementary material to expand the information. |
|
Are the results clearly presented? |
Can be improved |
We appreciate your suggestion and will address this issue further by improving the information in the figures. |
|
Are the conclusions supported by the results? |
Can be improved |
We appreciate your feedback. The conclusions are presented in a clear and concise manner and are fully supported by the research results. |
|
Are all figures and tables clear and well presented? |
Must be improved |
Thank you for your feedback. We have improved the presentation of figures and tables. |
|
3. Point-by-point response to Comments and Suggestions for Authors |
||
|
Comments 1: In the supplementary material, the scale bars in Figures S7 and S8 are completely indistinguishable. It is also barely visible in Figure 3 of the manuscript. |
||
|
Response 1: Your observation is important and we appreciate it. We have made the corresponding corrections to Figures S7 and S8 in the Supplementary Material. We have also improved the scale of Figure 3 in the manuscript. |
||
|
4. Response to Comments on the Quality of English Language |
||
|
Improved. |
||
|
Response: Thank you for your comment. |
||
Reviewer 2 Report
Comments and Suggestions for Authors
In the initial peer review, I judged that the manuscript could not be accepted for publication in Molecules, as it contained a total of 27 issues, including deficiencies in the English language. In the revised version, the authors have adequately addressed my comments and suggestions, and the manuscript has been substantially improved. I can now recommend it for publication in Molecules.
Author Response
|
Response to Reviewer 2 Comments
|
||
|
1. Summary |
|
|
|
Thank you very much for taking the time to review this manuscript. Please find the detailed responses below and the corresponding revisions/corrections highlighted/in track changes in the re-submitted files. |
||
|
2. Questions for General Evaluation |
Reviewer’s Evaluation |
Response and Revisions |
|
Does the introduction provide sufficient background and include all relevant references? |
Yes
|
We appreciate your opinion. |
|
Is the research design appropriate? |
Yes |
We appreciate your opinion. |
|
Are the methods adequately described? |
Yes |
We appreciate your opinion. |
|
Are the results clearly presented? |
Yes |
We appreciate your opinion |
|
Are the conclusions supported by the results? |
Yes |
We appreciate your opinion. |
|
Are all figures and tables clear and well presented? |
Yes |
We appreciate your opinion. |
|
3. Point-by-point response to Comments and Suggestions for Authors |
||
|
Comments 1: In the initial peer review, I judged that the manuscript could not be accepted for publication in Molecules, as it contained a total of 27 issues, including deficiencies in the English language. In the revised version, the authors have adequately addressed my comments and suggestions, and the manuscript has been substantially improved. I can now recommend it for publication in Molecules. |
||
|
Response 1: We appreciate your feedback and thank you for your acceptance. |
||
|
4. Response to Comments on the Quality of English Language |
||
|
The English is fine and does not require any improvement. |
||
|
Response: We appreciate your opinion. |
||